# MSF-ACA: Low-Light Image Enhancement Network Based on Multi-Scale Feature Fusion and Adaptive Contrast Adjustment

**DOI:** 10.3390/s25154789

**Published:** 2025-08-04

**Authors:** Zhesheng Cheng, Yingdan Wu, Fang Tian, Zaiwen Feng, Yan Li

**Affiliations:** 1School of Science, Hubei University of Technology, Wuhan 430068, China; 18302758631@163.com (Z.C.); 13437239003@163.com (Y.L.); 2College of Informatics, Huazhong Agricultural University, Wuhan 430070, China; fangzhihai_2003@mail.hzau.edu.cn (F.T.); zaiwen.feng@mail.hzau.edu.cn (Z.F.)

**Keywords:** low-light image enhancement, multi-scale fusion network, adaptive contrast enhancement, lightweight

## Abstract

To address the issues of loss of important detailed features, insufficient contrast enhancement, and high computational complexity in existing low-light image enhancing methodologies, this paper presents a low-light image enhancement network (MSF-ACA), which uses multi-scale feature fusion and adaptive contrast adjustment. Focus is placed on designing the local–global image feature fusion module (LG-IFFB) and the adaptive image contrast enhancement module (AICEB), in which the LG-IFFB adopts the local–global dual-branching structure to extract multi-scale image features, and utilizes the element-by-element multiplication method to fuse the local details with the global illumination distribution to alleviate the problem of serious loss of image details, while the AICEB incorporates linear contrast enhancement and confidence adaptive stopping mechanism, which dynamically adjusts the computational depth according to the confidence of the feature map, balancing the contrast enhancement and computational efficiency. According to the results of the experiment, the parameter count of MSF-ACA is 0.02 M, and compared with today’s mainstream algorithms, the suggested model attains 21.53 dB in PSNR when evaluated on the LOL-v2-real evaluation dataset, and the BRI is as low as 16.04 on the unpaired dataset DICM, which provides a better detail clarity and color fidelity in visual enhancement, and it is a highly efficient and robust low-light image model.

## 1. Introduction

Low-light images are generally caused by factors such as insufficient light given by the environment, limitations or malfunctions of the equipment, noisy signals, or improperly set parameters when shooting. The presence of such images can make it difficult to perform complex optical tasks, which include identity recognition and target detection. There is widespread use of low-light image enhancement algorithms in security surveillance, medical imaging, and other fields [1,2,3,4]. How can the luminance of an image be enhanced, its color recovered, and its texture details improved, while also boosting the computational efficiency of enhancing low-light images. Studying this area is essential for enhancing images taken in low light. Low-light image enhancement is one of the hot topics in the field of image processing.

Image enhancement algorithms can be classified into two main groups: conventional methods using manually constructed parameters and deep learning-based methods for improving images.

Development using traditional methods based on artificially constructed parameters began in the mid-to-late 20th century. These methods improve the overall visual effect by adjusting the illumination, contrast, and color of an image using artificial parameters. Typical algorithms include histogram equalization [5,6,7], gamma correction [8,9,10], and Retinex [11,12,13,14]. The histogram equalization method improves image contrast by redistributing values across pixels; the gamma correction method adjusts the luminosity and contrast of an overexposed or underexposed image using a non-linear transformation; and Retinex algorithm, which was proposed in collaboration with McCann and Land, is rooted in the visual system of the human eye and improves the effect of uneven lighting by separating the light and reflection components. Although these traditional algorithms can effectively improve image clarity and visual effect, they have problems such as relying on experts’ experience, poor scene adaptability, and general enhancement effect.

Deep learning-based image enhancement algorithms are gradually becoming mainstream methods as their results are generally better than traditional algorithms. According to the training method, it falls into two categories: supervised methods and unsupervised methods. Methodologies involving supervision necessitate the acquisition of paired datasets under low and normal lighting conditions through synthesis, shooting and data enhancement, and utilize the paired data for supervised training of image enhancement models. Common low-light image enhancement datasets include LOL dataset [15], MIT-Adobe FiveK dataset [16], LIME dataset [17], etc.

Zhang et al. [18] proposed Image Kindling the Darkness (Kindling the Darkness), which uses supervised learning to train the network using paired image datasets with good results. The network first breaks the image down into light and reflection parts. Then, the light component is used for lighting adjustments, while the reflection component is employed for removing degradations. However, it requires multi-stage training and the convolutional neural networks (CNNs) used to decompose color images perform noise reduction on reflectance, and need to be trained independently to adjust illumination and then connected together for end-to-end fine-tuning, which complicates and lengthens the process of training. As an extension of the Retinex framework, Wu et al. [19] put forward a deep unrolling network called URetinex-Net, which makes use of pair-wise data, and implements image enhancement through a manually designed a priori and optimization-driven approach. This method achieves excellent results in noise suppression and feature detail preservation. However, its strong dependence on training data and high computational complexity limit its generalization ability in complex unknown scenes. FMR-Net, proposed by Chen et al. [20], is a fast multiscale residual network which rapidly boosts the image quality in low-light by combining a highly optimized residual block with a multibranch structure, while preserving image details and contrast, maintaining image details and contrast. However, it suffers from serious loss of high-frequency details and insufficient noise suppression.

The above methods are mainly based on supervised learning. In order minimize dependency on paired illumination datasets, numerous scholars put forward image enhancement methods founded on unsupervised learning for low-light conditions. Jiang et al. [21] proposed EnlightenGAN, which uses generative adversarial networks for unsupervised learning and introduces a self-regularization mechanism. But it suffers from limited generalization and unstable training, etc. Guo et al. [22] proposed a zero-reference depth curve estimation method-Zero-DCE, which learns the dynamic range adjustment through image-specific curves, but its problem is that the accuracy is limited under extreme conditions and the non-reference loss function relies on the image-specific curves, but its accuracy is limited under extreme conditions and the non-reference loss function relies on the image-specific curves. The problem lies in the limited accuracy under extreme conditions and the dependence of the non-reference loss function on empirical parameters (e.g., exposure values). Ma et al. [23] proposed a self-calibrating illumination (SCI) learning framework, which achieves fast convergence of multi-stage outputs to a consistent state through a progressive illumination estimation module with shared weights and a self-calibration module, thus requiring only single-stage inference at test time. Combined with unsupervised training loss (fidelity and smoothing constraints), it enhances the model’s adaptability to complex low-light scenes. However, the network still suffers from detail loss or color bias.

Currently, low-light image enhancement algorithms primarily suffer from detail feature degradation, inadequate contrast enhancement, as well as the model has a high computing burden; for this reason, a low-light image enhancing network is put forward in this paper, which is founded upon multi-scale feature fusion and adaptive contrast adjustment, and designs a local–global image feature fusion module (LG-IFFB) and an adaptive image contrast enhancement module (AICEB). LG-IFFB extracts multi-scale image features through a local–global dual-branch structure and uses element-by-element multiplication to fuse the local details with the global light distribution to alleviate the problem of image detail loss; AICEB combines a linear contrast enhancement formula with a confidence stability-driven adaptive stopping mechanism to dynamically adjust the computational depth according to the confidence of the feature map, balancing the contrast enhancement and computational efficiency. As shown in Figure 1, the method proposed in this paper achieves excellent results when compared qualitatively and quantitatively with some of the current mainstream methods. The primary outcomes of this study are summarized below:

1.A low-light image enhancing network based upon multi-scale feature fusion and contrast adaptive adjustment is proposed to achieve low-light image enhancement through a lightweight architecture synergizing multi-scale image feature fusion with dynamic optimization of image contrast.2.A local–global image feature fusion module (LG-IFFB) is designed, which adopts a dual-path structure of local branching and global branching to simultaneously extract local and global information at different scales of the image, realizing a balance between detail preservation and global light optimization, and providing parameter mapping more suitable for complex low-light scenes for the subsequent luminance enhancement network.3.An adaptive image contrast enhancement module (AICEB) is proposed, which consists of multiple iterative sub-modules, each of which dynamically generates contrast enhancement factors and luminance parameters through an adaptive attention normalization block (AANBlock). A confidence scoring mechanism is introduced in the module to realize the adaptive contrast enhancement, effectively balancing the contrast enhancement and computational efficiency.

## 2. Methods

The MSF-ACA model architecture is illustrated in Figure 2. Firstly, the low-light image undergoes a convolutional operation that increases its amount of channels for feature maps, resulting in 24 channels. Then, using the local–global image feature fusion module (LG-IFFB), the multi-scale image features U are extracted and fused. In the luminance enhancement network, the multi-scale image features U are first uniformly partitioned along the channel dimension, and each group of the segmented features serves as a luminance tuning parameter mapping, which is utilized to perform luminance adjustment on the original image m. Specifically, through the optimization process of eight iterations, the output of the Luminance-enhanced feature map Θ, which will then be used as a component of the input to two subsequent cascaded Adaptive Image Contrast Boosting Modules (AICEBs), where the input to the first AICEB is the two Θs, and the output is the intermediate feature map Y. Input to the second AICEB is the intermediate feature maps Y and Θ, while its output represents the intermediate features map Z. AICEB dynamically generates contrast factors and luminance parameters, and adaptively terminates the redundant computation through a confidence scoring mechanism to balance performance and efficiency. The two AICEBs sequentially optimize the features in a recursive form to gradually improve the image contrast and brightness. Finally, MSF-ACA fuses the channels and reconstructs the image by using a 3 × 3 convolutional fusion channels. The main structure of MSF-ACA references the residual join learning mechanism of local combination global [25].

### 2.1. Local–Global Image Feature Fusion Block

Local–Global Image Feature Fusion Block (LG-IFFB): Figure 2 illustrates its structure; the division of the process is primarily into two steps: first, multi-scale feature extraction; second, image feature fusion.

Inspired by the structure of RFDB [25] as well as RRFB [25], LG-IFFB employs a multi-scale reparameterization technique for feature extraction, which outputs six different scales of image features (denoted as *L*1…*L*6) by cascading six kinds of convolutional kernel to maximize the preservation image texture information. In order to reduce memory loss, ADD operation is used instead of Concat operation for feature fusion. In addition, the remaining image feature channels are then directly combined with the fused features using the residual join method. This design is conducive to enhanced computational efficiency as well as reduced feature redundancy within the cascade structure.

Traditional multi-scale feature fusion methods [26] usually perform only simple feature splicing or summing processing, which cannot deeply fuse to utilize the global contextual information of the features. In order to realize the adaptive fusion of image features, LG-IFFB is designed with a two-branch structure: local branch and global branch. The local branch focuses on the optimal fusion of local features, and the usual practice is to extract local features using a single 3 × 3 convolution [27,28,29], which ignores the correlation of multi-scale local features. For this reason, in this paper, the input multiscale image features (*L*1…*L*6) undergo a 1 × 1 convolutional process that extends their dimensionality. Next, a deep convolutional process is employed to extract local information, and, in particular, for each scale, the extracted local features are spliced with other local features extracted at different scales to enhance the correlation. The final locally generated optimized features *L*1*^L^*…*L*6*^L^* is obtained by processing the spliced features using deep convolutions and activating functions. In the global branch, the input feature channels are segmented, with only one-fourth of the total number of channels selected to extract and fuse features. Specifically, to integrate multi-scale contextual information, the features at all scales (*L*1…*L*6) are first aggregated by element summation to generate the combined representation U′ [30]. The global feature descriptor S is then computed by performing global average pooling (GAP) on U′, which compresses the spatial dimensions to preserve channel statistics and ensures that the feature descriptor S encodes the overall illumination and structural patterns of the entire image. A fully connected (FC) network consisting of two linear layers, a ReLU activation layer, and a Sigmoid layer processes the global feature descriptor S to obtain the attentional weights S′. The attention weight S′ is calibrated by multiplying it with each scale feature to obtain the global calibration feature *L*1*^G^*…*L*6*^G^*. The global calibration features (*L*1*^G^*…*L*6*^G^*) captures long-range contextual correlations, which are then combined with local optimization features (*L*1*^L^*…*L*6*^L^*) are multiplied and fused element by element to generate the final fused feature map U. This fusion strategy ensures that the final output U contains both fine local details and globally consistent illumination information, thus addressing the limitation of the single-branch approach that is unable to extract both local and global features simultaneously. Compared with the DCE-Net [22] in Zero-DCE, which relies primarily on single-branch convolution for learning the mapping relating the input image to the best-fit curve, LG-IFFB can address the issue of detail loss or contrast imbalance in extremely low-light or high-dynamic-range scenarios more effectively by integrating the link with multi-scale local details and global light distribution.

### 2.2. Luminance Enhancement Network

This study proposes a luminance enhancement function for the luminance Enhancement Network based on the illumination adjustment curves framework of Zero-DCE. The developed recurrent mapping function adheres to two fundamental design criteria: (1) the preservation of inter-pixel intensity relationships through enforced monotonicity constraints, and (2) the implementation of computational simplicity and gradient accessibility to enable efficient error backpropagation. The luminance enhancement function is formulated as a parametric quadratic transformation and can be represented mathematically by the following expression:(1)$$E_{p}\left(m\right)=E_{p-1}\left(m\right)+S_{p}\left(m\right)E_{p-1}\left(m\right)\left(1-E_{p-1}\left(m\right)\right),p\leq 8$$
where *S_p_*(*m*) is the pixel-by-pixel parameter matrix (*p* denotes the number of iterations and *m* denotes the input image), the luminance-enhanced image resulting from *p* iterations is known as *E_p_*(*m*), whereas the enhanced version of this image after *p* − 1 iterations is referred to as *E_p_*_−1_(*m*). Based on (1), this paper presents a design for a luminance enhancement network; refer to Figure 2. Two parts make up the network’s inputs: the original low-light image *m* ∈ *R^H^*^×*W*×3^ and the fused feature map U ∈ *R^H^*^×*W*×24^ after LG-IFFB processing. The final output of the luminance feature map Θ ∈ *R^H^*^×*W×*24^ with a channel number of 24 is obtained by dividing the LG-IFFB output of the fused feature map. U is partitioned into 8 identical pixel-by-pixel parameter maps (*S_p_*(*m*), *m =* 1, …, 8) in order to take part in the iteration of the function. The initial input original image *E*_0_(*m*) *= m*, after 8 iterations, outputs a 3-channel enhanced image *E*_8_(*m*), which is subsequently extended to 24 channels by 3 × 3 convolution to generate the luminance-enhanced feature map Θ = Conv(*E*_8_(*m*)).

### 2.3. Adaptive Image Contrast Enhancement Block

Direct contrast enhancement of input features (e.g., linear stretching) introduces noise or distortion due to unstable feature distribution. In a style migration task, Vedaldi et al. [31] argued that instance normalization techniques can eliminate the contrast difference in image features in order to allow the network to focus on learning content structure rather than luminance or color distribution. Drawing on this idea, this paper uses the channel normalization technique to improve the stability of contrast enhancement of image features. The channel normalization operation is to normalize each channel of the input feature map separately so that its mean is 0 and variance is 1, i.e.,(2)$$\overset{\wedge }{h}=\frac{h_{c}-\mu_{c}}{\sqrt{\sigma_{c}^{2}+\varepsilon }}$$
where the feature map of channel *c* is represented by *h_c_*, *μ_c_* refers to the mean values of channel *c*, and *σ_c_* refers to the standard deviation value of channel *c*. *ε* is a stabilization factor to avoid a zero denominator.

On this basis, this paper proposes an adaptive image contrast enhancement block (AICEB). Figure 3a illustrates the framework of the AICEB. The AICEB consists of several iterative submodules (Iteration Sub-module), and each Iteration Sub-module contains 1 Adaptive Attention Normalization Block (AANBlock) and 1 ReLU Activation layer. Figure 3b demonstrates the detailed architecture of the AANBlock.

The AANBlock input in the kth submodule has two inputs: the luminance enhancement feature map Θ obtained after processing by the luminance enhancement network and the preceding iterative submodule *X_k_*’s output feature map. For the 1st iterative submodule, *X*_1_ is the luminance feature map produced from the luminance enhancement network, which is represented by the symbol Θ. Through the learning of the luminance enhancement feature map Θ, it generates the parameters *a* and *b* needed for image contrast adjustment. Thus it allows the network to adaptively adjust the image contrast according to the luminance distribution information adaptively. The specific realization process is as follows:

In each iterative sub-module, take the kth iterative sub-module as an example, the features of the luminance feature map Θ are first further extracted using a 3 × 3 convolution and a 5 × 5 convolution to obtain the feature map Θ’. Then, the global information of Θ’ is computed by Global Average Pooling (GAP) and Global Maximum Pooling (GMP), respectively, and the outputs of GAP and GMP are merged in the dimension of the channel to obtain statistical features of global information. The global information statistical features obtained from splicing are processed by a Fully Connected (FC) Layer and Rectified Linear Unit (ReLU) activation functions are applied to generate the Channel Attention Weight T. Multiplying Θ’ with the channel attention weight T produces the feature map *α**. The ReLU Activation Function is used for guaranteeing that pixel values *α** are non-negative. Based on *α**, a 7 × 7 convolutional layer combined with a Sigmoid Activation Function needs to be used in order to generate the feature map *β**. In the formula for linearly enhancing the contrast, pixel values from the feature maps *α** and β* act as the parameters *a* and *b*.

Employing parameters *a* and *b* derived from feature maps *α** and *β**, a linear contrast transform is applied to the features that have been normalized by the channels, and the contrast-enhanced result *X*’*_k_*_+1_ can be obtained. The output after ReLU activation processing is then output, i.e., it is the output of the kth iterative sub-module, *X_k_*_+1_. Particularly, in order to ensure the effective enhancement of the low-light image contrast, a bias term of magnitude 1 is added to *α** is added with a bias term of size 1, i.e., *α** + 1. The specific formulation is set out below:(3)U(m,n)=a×T(m,n)+b,0<m<H,0<n<W
where T(m,n) and U(m,n) denote the image features to be enhanced and the image features after contrast enhancement, respectively; *a* and *b* are the parameters of image contrast enhancement, specifically the pixel values of the feature maps *α** and *β**.(4)$$X_{k+1}=\mathrm{Relu}((\alpha^{*}+1)\frac{X_{k}-\mu }{\sigma }+\beta^{*})$$
where *σ* and *µ* represent the standard deviation and average of the feature map *X_k_*, respectively.

In order to balance the image contrast enhancement effect and computational efficiency, this paper sets out a proposal for an adaptive stopping mechanism founded on the stability of the confidence level. While iteratively enhancing the image contrast, the contrast confidence level is calculated in real time to evaluate the enhancement results, and the computational depth of the network is adaptively adjusted, and its flow is presented in Figure 3a.

After the iterative sub-module completes the image contrast enhancement processing, the confidence of the current results is generated using the contrast confidence calculation module C. The contrast confidence calculation module C consists of a variance calculation layer (Contrast), a Layer 1 × 1 Convolution (Conv) followed by a Sigmoid Activation layer. Confidence is calculated as follows:(5)Confidence=Sigmoid(Conv(Contrast(x)))∈[0,1]
where the variance calculation layer Contrast(*x*) is(6)$$Contrast(x)=\frac{1}{hw}\sum_{u,v}(x_{u,v}-\mu {)}^{2}$$*w* and *h* refer to the width and height of the feature map *x*; *x_u_*_,_*_v_* represent a pixel’s value at position (*u*, *v*) in feature map *x*; and *μ* indicates the mean value of all pixels in *x*. Variance is widely used to measure image pixel dispersion; the higher the variance, the greater the difference between pixels, i.e., the higher the contrast. If the absolute value of the difference in the confidence level for three consecutive times is less than the preset threshold *λ*, the enhancement effect of the feature map can be considered to be stabilized, and the iteration is terminated and exited:(7)$$\left|Confidence_{k}-Confidence_{k-1}\right|<\lambda ,\forall k\in \{f,f-1,f-2\}$$
where Confidence*_k_* represents the confidence at the kth layer, *f* is the index of the current iteration number *k*, and the value of *f* is not less than 3. The preset threshold *λ* is obtained by experiments on the validation set [15] by taking into account the average number of iterations and the PSNR loss, which is specifically taken as 0.0005.

### 2.4. Loss Function

For the purpose of optimizing MSF-ACA training, this paper uses spatial consistency loss, color consistency loss, mean absolute error loss, gradient-guided structural consistency loss, and perceptual loss to comprehensively evaluate the image enhancement effect.

**Spatial Consistency Loss (*L_spa_*):** The structural consistency before and after enhancement is maintained by analyzing the luminance difference in the local region of the image. Firstly, the image is divided into *M* pixel blocks, where each block *p* is compared with its four neighborhoods *Φ*(*p*). Then the luminance differences between the reference image *S* and the enhanced image *T* between the corresponding blocks are calculated, respectively, and finally the stability of the spatial relation is constrained by the mean value of the squared error:(8)$$L_{spa}=\frac{1}{M}\sum M_{p=1}^{M}\sum_{q\in \Phi (p)}{\left(\left|S_{p}-S_{q}\right|-\left|T_{p}-T_{q}\right|\right)}^{2}$$

**Color Consistency Loss (*L_col_*):** It is used to ensure the stability of the color distribution of the content during the image enhancement process. The loss function achieves this goal by constraining the intensity differences between different color channels. For any pair of channels (*m*, *n*) in the set of channel pairs *ε*, the sum of the squared differences in their average intensity values $I^{m}$ and $I^{n}$ is computed:(9)$$L_{col}=\sum_{\forall (m,n)\in \varepsilon }{\left(I^{m}-I^{n}\right)}^{2},\varepsilon =\left\{\left(R,G),(R,B),(G\times B\right)\right\}$$

**Mean Absolute Error Loss (*L*_1_):** It is mainly used to prevent images from localized overexposure or underexposure. This loss function achieves this goal by comparing the difference in pixel luminance between the reference image and the enhanced image. The luminance value of the ith pixel in the reference image is denoted as *Y_i_*, and the luminance value of the corresponding pixel in the enhanced image is denoted as *O_i_*, and let the total number of pixels in the image be *N*:(10)$$L_{1}=\frac{\sum_{i=1}^{N}\left|Y_{i}-O_{i}\right|}{N}$$

**Gradient-guided Structural Consistency Loss (*L_gsc_*):** By establishing structural similarity constraints in the gradient domain, the consistency of luminance, contrast, and structural features during image enhancement is effectively maintained. This paper presents the gradient-based structural similarity loss function (*G_SSIM_*). The *G_SSIM_* method outperforms the conventional SSIM method in images with low light and blurring [32]. The gradient magnitude of the augmented image, *O*, is expressed as *G_o_*(*u*, *v*), whilst the gradient magnitude of the reference images, *Y*, is expressed as *G_Y_*(*u*, *v*). (*u*, *v*) are the row-column coordinates representing the pixels in the image, whereas *C* acts as a stabilization constant to prevent the denominator from equaling zero:(11)$$L_{gsc}=1-\frac{2\sum_{u}\sum_{v}G_{o}(u,v)G_{Y}(u,v)+C}{\sum_{u}\sum_{v}{\left[G_{o}(u,v)\right]}^{2}+\sum_{u}\sum_{v}{\left[G_{Y}(u,v)\right]}^{2}+C}$$

**Perceptual Loss (*L_perceptual_*):** Perceptual loss aims to maintain semantic consistency between the augmented image and the reference image through deep feature alignment. To achieve this goal, this study uses an ImageNet-based pre-trained VGG network architecture as a static feature encoder. The layer *l* features obtained from the pre-trained VGG network are represented by *Φ_l_*(*x*) and *Φ_l_*(*y*). *x* refers to the low-light-enhanced output image, *y* denotes the reference image and λ*_l_* indicates the weight used for the perceptual loss of layer *l*. The VGG network is used to minimize the distance between the low-light-enhanced image and reference image. ‖.‖_2_ denotes the *L*2 paradigm.(12)$$L_{perceptual}(x,y)={\sum_{l}\lambda_{l}\left\|\Phi_{l}(x)-\Phi_{l}(y)\right\|}_{2}^{2}$$

The total loss function (*L_total_*) is the weighted average of the above five loss functions.(13)$$L_{total}=\omega_{1}L_{spa}+\omega_{2}L_{col}+\omega_{3}L_{1}+\omega_{4}L_{gsc}+\omega_{5}L_{perceptual}$$
where $\omega_{1}\ldots \ldots \omega_{5}$ is the weight parameter of each loss function.

## 3. Experiment

### 3.1. Implementation Details

In this paper, a total of 1385 images are selected from the LOL datasets series (v1 [15] and v2 [33]), and with the objective of speeding up the training process, a 256 × 256 pixel size window is selected, and the input images undergo center clipping. In the course of network training, Adam [34] optimizer was used. Adam optimizer optimized the network for 40 cycles (batch size = 1, learning rate = 10^−4^). Initialization of all the network weights was performed using a zero-mean, 0.02 standard Gaussian function, while the bias was originally defined as a fixed value. Implementation of this entire process relied on the PyTorch 1.13.0 [35] framework, and using an NVIDIA GeForce RTX 4060 graphics card, MSF-ACA underwent both training and evaluation phases.

To evaluate the proposed low-light image enhancement approach, the proposed method is compared with eight current state-of-the-art low-light image enhancement algorithms, specifically, CLIP-LIT [36], DSLR [37], SCI [23], RUAS [38], URetinex [19], HEP [24], DeepLPF [39], and FMR-Net [20]. It is worth noting that some of these methods (e.g., HEP and RUAS) are unsupervised frameworks. According to [40], PSNR and SSIM are used to assess the effect of contrast enhancement. PSNR focuses on pixel differences, while SSIM evaluates visual structural similarity.

### 3.2. Comparison and Analysis of Paired Datasets

**Quantitative Analysis:** The efficacy of different enhancement methods on LOL-v2 subsets (LOL-v2-real and LOL-v2-syn) was assessed through four metrics: PSNR and SSIM [41] for quality evaluation, complemented by FLOPs for computational efficiency analysis, and the number of model parameters (Params) were chosen to analyze the computational complexity of the model. As can be seen from Table 1, excellent results were achieved in regard to the PSNR, FLOPs, and parametric evaluations are conducted across both real-world and synthetic versions of the LOL-v2 benchmark. Larger values of the PSNR and SSIM metrics represent higher image quality and are indicated by ‘↑’ in Table 1. SCI along with RUAS show remarkable model lightweighting performance, but the PSNR and SSIM of the LOL-v2-real and LOL-v2-syn datasets are significantly lower than those of the MSF-ACA. On the LOL-v2-real dataset, MSF-ACA has a PSNR of 21.53 dB, which is better than URetinex’s 21.09 dB, and an SSIM of 0.771, which is slightly lower than that of URetinex’s 0.858. On the LOL-v2-syn dataset, MSF-ACA ranks higher in both PSNR and SSIM and is ranked first in both metrics. With regard to the two important performance metrics, PSNR and SSIM, which characterize an image enhancement effect, both achieved the top two places, and the FLOPs of MSF-ACA of this paper’s method is 29.97 G, which is higher than that of URetinex’s 1.24 G. However, the comprehensive performance of MSF-ACA is better than that of URetinex on both datasets, with the performance gain far outweighing the increase in the computing cost. The size of the parameter of both models is 0.02 M, which fully verifies the balance between performance and efficiency in lightweight design, and achieves SOTA effect in image quality and model complexity.

**Visual Analysis:** Figure 4 and Figure 5 show the visual comparison results between this paper’s method and other state-of-the-art image enhancement methods on paired datasets. The LOL-v2-real dataset is selected in Figure 4 and the LOL-v2-Synthetic dataset is selected in Figure 5. The results from Figure 4 and Figure 5 show that the FMR-Net method is poor in detail retention, although it improves in brightness, with obvious noise and artifacts in some regions. DeepLPF is deficient in both detail retention and brightness enhancement. On the other hand, the method in this paper not only significantly improves the overall brightness when enhancing the low-light image, but also performs well in detail retention; the details in the image are clearly visible, yielding an effect that matches the ground truth in visual fidelity. In addition, as can be seen from the zoomed-in regions in Figure 4 and Figure 5, FMR-Net still lacks in the processing of details, with obvious noise appearing in some regions. URetinex has the problem of color deviation on LOL-v2-Synthetic, and the color and luminance of the enhanced image produced by DeepLPF deviates significantly from the referenced image, compared to which, the approach outlined presented in this paper effectively boosts local and global contrast, producing sharper details while also performing well in noise suppression for better visual results.

**Figure 4 sensors-25-04789-f004:**
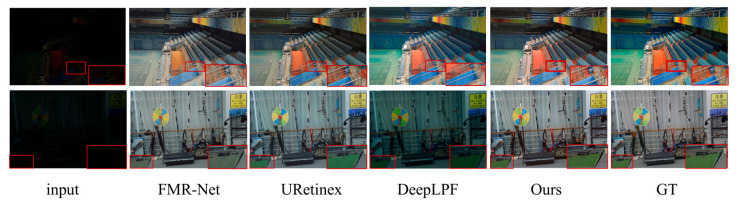
Visual comparison on the LOL-v2 real dataset. There are two red boxes in the figure, the larger red box is an enlargement of the smaller red box. Same as Figure 5, Figure 6 and Figure 7.

**Figure 5 sensors-25-04789-f005:**
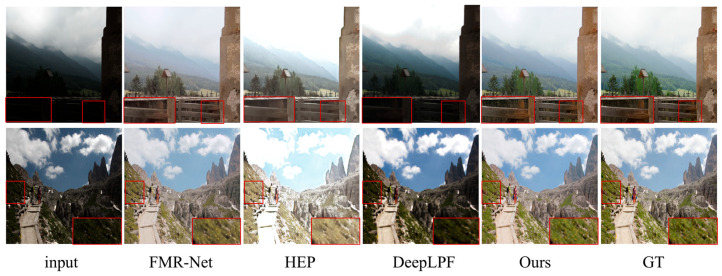
Visual comparison on the LOL-v2 Synthetic dataset.

**Figure 6 sensors-25-04789-f006:**
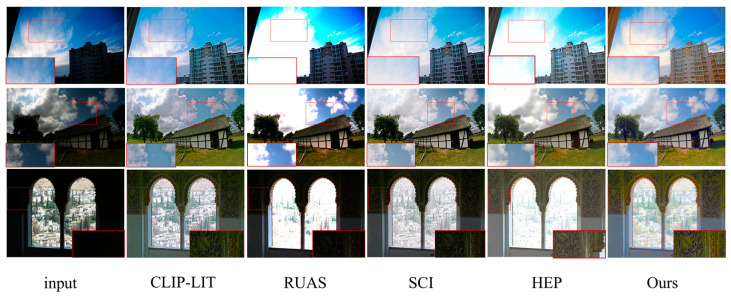
Visual comparison on DICM, MEF, and VV non-reference datasets.

**Figure 7 sensors-25-04789-f007:**
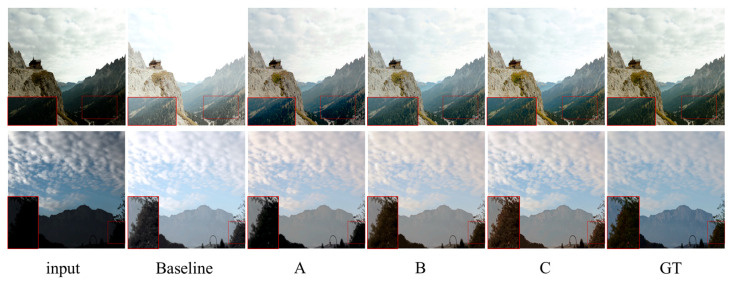
Visual comparison of ablation studies.

### 3.3. Comparison and Analysis of Unpaired Datasets

**Quantitative Analysis:** The proposed MSF-ACA was compared with competing methodologies using five unpaired datasets (LIME [7], DICM [10], NPE [42], MEF [43], and VV) to further validate its effectiveness. Two non-reference perceptual metrics, PI [44] and BRI [45], were chosen for the evaluation of the visual quality of the enhancement results. The visual quality is indicated by lower metrics. Smaller values of the BRI and PI metrics represent higher image quality, as indicated by ‘↓’ in Table 2. As demonstrated in Table 2, experimental results demonstrate that our approach obtains the lowest BRI and PI scores on LIME, DICM, and MEF datasets, as well as the lowest PI scores across the NPE and VV datasets. In the context of average scores across the two metrics, the method in this paper is the lowest on all five datasets, indicating the proposed method still achieves satisfactory performance metrics in unseen real-world scenarios.

**Visual Analysis:** Figure 6 shows the visual comparison results between this paper’s method and the competing methods on the unpaired dataset, and rows 1–3 in the figure correspond to the image enhancement results from the DICM, MEF, and VV datasets, respectively. For images from the DICM dataset, the overall brightness of the house and the details of the sky sunset color can be better recovered; for images from the MEF dataset, the sky and cloud colors can be enhanced more naturally; for images from the VV dataset, the RUAS method and the SCI method underperform in the recovery of the overall brightness, and in comparison with the CLIP-LIT and the HEP methods; the method in this paper can better recover the texture color of the wall. Overall, this paper’s method can still effectively recover the lighting and recreate the image’s details in unknown scenes, demonstrating its ability to generalize.

### 3.4. Ablation Experiment

The LOL-v2 dataset is used to perform ablation experiments. Using the luminance-only enhancement network as the baseline model, the performance difference is analyzed by adding the corresponding modules to verify the efficacy of the Local–Global Image Feature Fusion Module (LG-IFFB), and Adaptive Image Contrast Enhancement Block (AICEB) proposed in this paper.

**Local–Global Image Feature Fusion Block:** Model A introduces only the local–global image feature fusion module based on the baseline model, which improves the SSIM and PSNR metrics to 0.01 and 0.74 dB, respectively. From Figure 7, it is clear that model A results in a greater image texture than the baseline model, such as the clouds in the sky, the texture of trees, and so on. This proves that adding the local–global image feature fusion module can learn more expressive features.

**Adaptive Image Contrast Enhancement Block:** Model B and Model C are based on Model A, further adding Adaptive Image Contrast Enhancement Module (AICEB). A fixed number of iterations is used by Model B. In this paper, the iteration count for each AICEB is pre-set to 10 and there are two AICEB modules, so the overall iteration count is 20. Model C employs the adaptive stopping mechanism proposed in this paper.

From Table 3, it can be observed that there is no improvement in the SSIM metric value and the PSNR metric is improved by 0.39 dB for Model B compared to Model A. In Model C, by introducing the adaptive stopping mechanism, the number of iterations decreases by an average of 9.9 iterations and the average running time decreases by 0.08 s compared to Model B, while the PSNR and SSIM metric values are improved by 0.77 dB and 0.04, respectively. The accuracy and computational efficiency of the model in processing images are improved at the same time. From the visual test in Figure 7, the color and contrast of the image are significantly improved. The effectiveness of the AICEB module and the adaptive stopping mechanism proposed herein has been proven.

Ablation test outcomes for the quantity of AICEB modules, as carried out in this paper, are displayed in Table 4. When two AICEB modules are introduced into the model, compared with the introduction of two AICEBs, there is no improvement in either the PSNR or the SSIM metrics, but average iteration and running times have increased significantly. When the number of AICEB modules is two, a balance between the accuracy and efficiency of the model can be achieved.

### 3.5. Selection of Iteration Thresholds

According to [46], PSNR is used to evaluate the effect of contrast enhancement. So, the threshold *λ* of the adaptive stopping mechanism in the AICEB module of this paper is determined based on a combination of the average number of iterations and PSNR loss. From 0.00001 to 0.001, five different thresholds are selected, and the number of iterations and PSNR values in different cases are shown in Table 5. It is not difficult to find that when *λ* = 0.0005, the model reduces the number of iterations by 14% while the PSNR loss is only 10%, which realizes the balance between accuracy and efficiency, and for this reason, the *λ* threshold is set to 0.0005 in this paper.

## 4. Conclusions

In this paper, a lightweight network (MSF-ACA) combining multi-scale feature fusion and contrast adaptive adjustment is proposed to effectively realize high-quality enhancement to images taken in low-light conditions. By designing a local–global two-branch feature fusion module (LG-IFFB), the integration of multi-scale local details and global illumination information effectively alleviates the problem of texture loss; an adaptive image contrast enhancement module (AICEB) is introduced to adaptively balance the computational efficiency and enhancement effect through a confidence scoring mechanism. A large number of experiments show that MSF-ACA not only outperforms existing methods on typical datasets but also has a significant advantage in computational complexity.

## Figures and Tables

**Figure 1 sensors-25-04789-f001:**
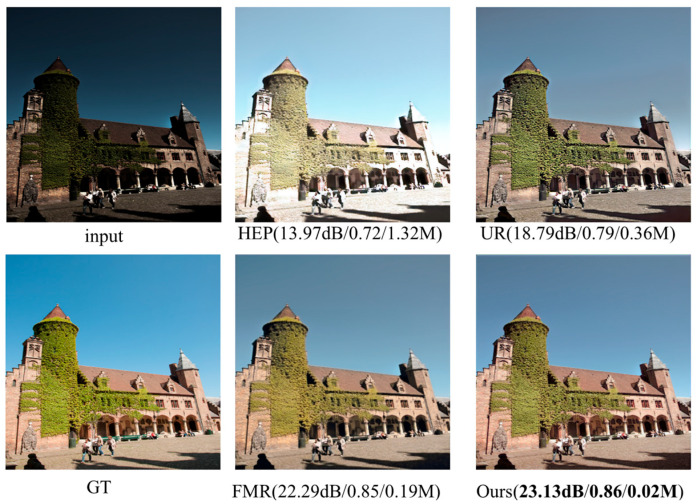
A comprehensive investigation into the enhancement of micro-optical images, encompassing state-of-the-art methodologies incorporating HEP [24], URetinex [19], and FMR-Net [20], is undertaken through a meticulous quantitative and qualitative analysis (i.e., PSNR/SSIM/parameter). Compared with other similar methods, the method in this paper achieves better enhancement results with fewer model parameters.

**Figure 2 sensors-25-04789-f002:**
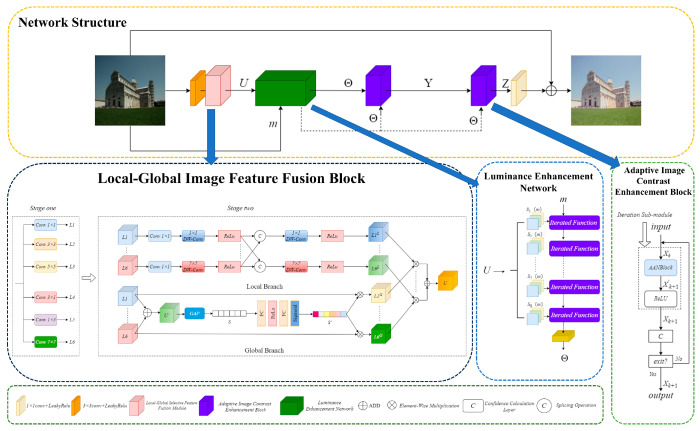
MSF-ACA model structure.

**Figure 3 sensors-25-04789-f003:**
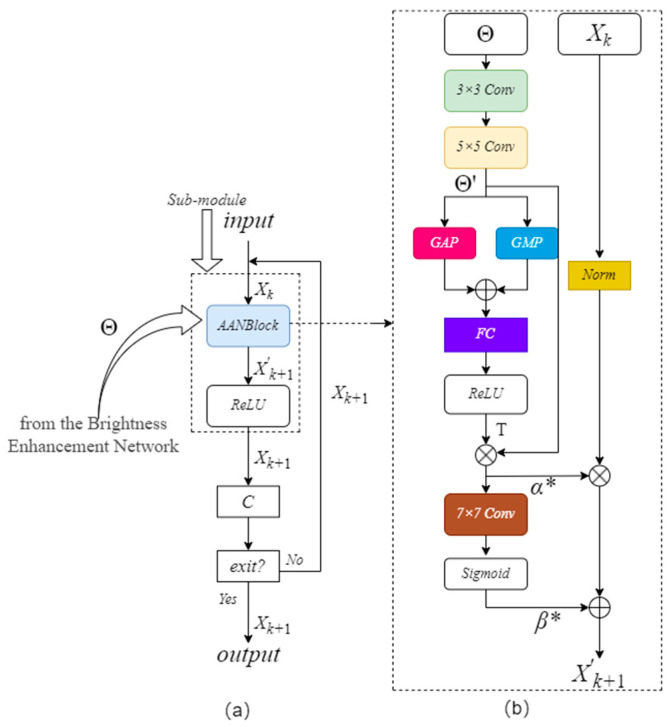
(**a**) Structure of AICEB; (**b**) structure of AANBlock.

**Table 1 sensors-25-04789-t001:** The LOL-v2 dataset is used to facilitate quantitative comparisons.

Methods	Complexity		LOLV2-Real		LOLV2-Syn	
	FLOPs (G)	Params (M)	PSNR ↑	SSIM ↑	PSNR ↑	SSIM ↑
CLIP-LIT	18.24	0.27	15.26	0.601	16.16	0.666
DSLR	5.88	14.93	17.00	0.596	13.67	0.623
SCI	0.06	0.0003	17.30	0.540	16.54	0.614
RUAS	0.83	0.003	18.37	0.723	16.55	0.652
URetinex	1.24	0.02	21.09	0.858	13.10	0.642
HEP	14.07	1.32	18.29	0.747	16.49	0.649
DeepLPF	5.86	1.77	14.10	0.480	16.02	0.587
FMR-Net	102.77	0.19	20.56	0.736	19.09	0.657
Ours	29.97	0.02	21.53	0.771	20.27	0.716

**Table 2 sensors-25-04789-t002:** Comparison of quantitative assessment of unpaired datasets.

Methods	DICM		LIME		MEF		NPE		VV	
	BRI ↓	PI ↓	BRI ↓	PI ↓	BRI ↓	PI ↓	BRI ↓	PI ↓	BRI ↓	PI ↓
CLIP-LIT	24.18	3.55	20.43	3.07	20.67	3.11	19.37	2.91	36.00	5.40
DSLR	25.67	4.07	22.68	6.01	22.49	6.74	33.69	5.07	28.35	6.64
SCI	27.92	3.70	25.17	3.37	26.71	3.28	28.88	3.53	22.80	3.64
RUAS	46.88	5.70	34.88	4.58	42.12	4.92	48.97	5.65	35.88	4.32
URetinex	24.54	3.56	29.02	3.71	34.72	3.66	26.09	3.15	22.45	2.89
HEP	25.74	3.01	31.86	5.74	30.38	3.28	29.73	2.36	39.86	2.98
DeepLPF	19.93	3.59	24.70	4.45	22.40	4.04	17.09	3.08	23.75	4.28
FMR-Net	19.63	2.91	28.96	3.77	21.67	3.25	18.01	2.70	17.56	2.64
Ours	14.45	2.36	16.61	2.76	18.31	2.70	25.44	1.78	28.02	2.32

**Table 3 sensors-25-04789-t003:** Results of ablation experiments. ‘↓’ means that the higher the value, the worse the results reflected by the corresponding indicator.

Models	LG-IFFB	AICEB (Fixed Iteration)	AICEB	PSNR	SSIM	Number of Iterations ↓	Time(s) ↓
Baseline				18.37	0.66	-	-
A	**√**			19.11	0.67	-	-
B	**√**	**√**		19.50	0.67	20	0.22
C	**√**		**√**	**20.27**	**0.71**	**10.1**	**0.14**

**Table 4 sensors-25-04789-t004:** Ablation studies of the number of AICEB modules.

Number of AICEB	1	2	3
PSNR/SSIM	19.79/0.68	**20.27 (+2.4%)/0.71 (+4.4%)**	20.20 (+2.0%)/0.71 (+4.4%)
Number of iterations	7.3	**10.1 (+38%)**	26.4 (+261%)
Average running time(s)	0.11	**0.14 (+27%)**	0.22 (+100%)

**Table 5 sensors-25-04789-t005:** Comparison of different thresholds and number of iterations and PSNRs.

Threshold	0.00001	0.00005	0.0001	0.0005	0.001
Number of iterations	20	19.5	19 (−3%)	**16.3 (−14%)**	15.1 (−20%)
PSNR	7.6	9.9	9.4 (−5%)	**8.9 (−10%)**	8.4 (−15%)

## Data Availability

Data are contained within the article.

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
