# Peer review of "MSF-ACA: Low-Light Image Enhancement Network Based on Multi-Scale Feature Fusion and Adaptive Contrast Adjustment"

_sensors, 2025, doi:10.3390/s25154789_

Round 1
Reviewer 1 Report
Comments and Suggestions for Authors
The manuscript presents a low-light enhancement network using multi-scale feature fusion along with an adaptive contrast adjustment. The methodology section is detailed and provided with good illustration to highlight the different parts of methodology. The experiments section contains comparison with some of the State-of-the-arts (SOTA) methods on LOL dataset as well as other datasets. The manuscript is overall well-written and shows good contribution.
However, some concerns need to be addressed by the authors before the manuscript can be accepted.
(a) The contrast adjustment block needs to be explained a bit more, since it is one of the major contributions. Currently, it is hard to understand how PSNR loss can provide a reliable measure for contrast. Some explanation or references would benefit the user.
(b) URetinex numbers in Table 1 look wrong. Even if we separate synthetic and real sub data, the maximum value of PSNR is 20.79 which is lower than the ones reported in their paper, or competitor papers [1]. Note that if we average or take maximum of any of the real or syn value from this manuscript, we cannot get the values reported in other papers. Please also check other values because the reviewers may check them again in a revision. Explanation is expected from the authors.
(c) Methods like HEP are unsupervised. Hence, when comparing it is better to mention that a supervised method is being compared to an unsupervised method. Supervision always has advantage over unsupervised ones. It is transparent to mention this. Please go through the reference [1] provided below to understand. The authors are also strongly advised to compare to the following reference.
[1] Bai, Jiesong, et al. "Retinexmamba: Retinex-based mamba for low-light image enhancement." International Conference on Neural Information Processing. Springer, Singapore, 2025.
Author Response
Comments 1:
The contrast adjustment block needs to be explained a bit more, since it is one of the major contributions. Currently, it is hard to understand how PSNR loss can provide a reliable measure for contrast. Some explanation or references would benefit the user.
Response 1:
Literature [1] uses PSNR to evaluate the effect of contrast enhancement, so this paper uses PSNR for evaluation.
[1] Talab, A. W., Younis, N. K., & Ahmed, M. R. (2024). Analysis Equalization Images Contrast Enhancement and Performance Measurement. Open Access Library Journal, 11(4), 1-11.
Comments 2:
URetinex numbers in Table 1 look wrong. Even if we separate synthetic and real sub data, the maximum value of PSNR is 20.79 which is lower than the ones reported in their paper, or competitor papers [1]. Note that if we average or take maximum of any of the real or syn value from this manuscript, we cannot get the values reported in other papers. Please also check other values because the reviewers may check them again in a revision. Explanation is expected from the authors.
Response 2:
Thank you for your careful examination of Table 1. Upon rechecking, we found that the URetinex results originally reported in our manuscript were different from those in [2]. We have updated the URetinex values in Table 1 based on the consistent results reported in [3] and [4], both of which use the official codebase and dataset splits provided by the original URetinex authors. The results in [2] may differ due to implementation details or preprocessing differences. Other metrics in Table 1 are checked carefully to make sure they are error-free.
For the method of Retinexmamba, which obtain a good performance in low-light image enhancement, due to the complexity of its model structure and the additional training and parameterisation work required, we are unable to achieve a fair comparison with this method in this paper for the time being. We have made comparison with eight current state-of-the-art low-light image enhancement algorithms, CLIP-LIT, DSLR, SCI, RUAS , URetinex , HEP , DeepLPF and FMR- Net , and the comparative experiments have been conducted in sufficient detail. Regarding the research on Retinexmamba, we will conduct an in-depth study on the ideas and strategies of this algorithm in the later stage, aiming to better improve the algorithm proposed in our paper.
[2] Bai, Jiesong, et al. "Retinexmamba: Retinex-based mamba for low-light image enhancement." International Conference on Neural Information Processing. Springer, Singapore, 2025.
[3] Xue, Minglong, et al. "Low-light image enhancement via clip-fourier guided wavelet diffusion." arXiv preprint arXiv:2401.03788 (2024).
[4] Jiang, Hai, et al. "Low-light image enhancement with wavelet-based diffusion models." ACM Transactions on Graphics (TOG) 42.6 (2023): 1-14.
Comments 3:
Methods like HEP are unsupervised. Hence, when comparing it is better to mention that a supervised method is being compared to an unsupervised method. Supervision always has advantage over unsupervised ones. It is transparent to mention this. Please go through the reference [2] provided below to understand. The authors are also strongly advised to compare to the following reference.
Response 3:
We agree with your comments. We make a note in Section 3.1 to explicitly point out this difference and add [2] to the references of the manuscript.
Reviewer 2 Report
Comments and Suggestions for Authors
The authors of the article ‘MSF-ACA:Low-Brightness Image Enhancement Network Based on Multiscale Feature Fusion and Adaptive Contrast Adjustment’ present a low-brightness image enhancement network consisting of three modules:
-the first module is called LL-IFFB which has the task of extracting multiscale U features from the images.
-The second module LEN has the purpose of improving the brightness of the initial image using the information present in the multiscale U features.
-The next last module is AICEB, which aims to improve contrast in the image.
The overall network was trained using the LOL dataset and the performance compared with other algorithms for improving low brightness images.
The description of the modules should be improved as it is unclear, especially section 2.1 describing the LL-IFFB. In particular, the description of the global branch would be improved by adding a reference to the aritcole "Learning Enriched Features for Fast Image
Restoration and Enhancement" S.W. Zamir et al.IEEE PAMI 45.2,2023.
The results obtained are very good and the authors also performed an analysis of the improvements of the various modules using the LEN module as a baseline.
There are several typographical errors in the article that require more careful revision by the authors.
Comments on the Quality of English Language line 19: there is a repetition "...the AICEB fuses the linear contrast The AICEB incorporates linear contrast..." the sentence should be "...the AICEB incorporates linear contrast..." lines 49-51: the sentence "Image enhancement algorithms can be classified into two main subsets: conventional methods that are manually constructed parameters, and deep learning-based methods for enhancing images." is not clear. I propose "Image enhancement algorithms can be classified into two main groups: conventional methods using manually constructed parameters and deep learning-based methods for improving images." line 57 : please change "...The gamma..." in "...the gamma..." line 142: please remove the first "image" word in the sentence line 143-144: what is the meaning of "the multi-scale image features U are first uniformly channelized", please clarify. line 147: there is a repetition of feature map symbol lines 169-170: I suggest to move the sentence "we segment the input feature channels and select only one-fourth of the total number of channels to extract and fuse features." in the section describing the global branch line 189: there is a repetition of "The global feature", please remove it. line 241-244: the sentence is a repetition of the previous sentence.Author Response
Comments 1:
The description of the modules should be improved as it is unclear, especially section 2.1 describing the LL-IFFB. In particular, the description of the global branch would be improved by adding a reference to the aritcole "Learning Enriched Features for Fast Image
Restoration and Enhancement" S.W. Zamir et al.IEEE PAMI 45.2,2023.
Response 1:
We agree with your comments. In order to express the global branching in the IG-IFFB module proposed in this paper more clearly, based on your suggestion, we refer to the article Learning Enriched Features for Fast Image Restoration and Enhancement and add the description of each component of the global branching in Section 2.1..
Comments 2:
line 19: there is a repetition "...the AICEB fuses the linear contrast The AICEB incorporates linear contrast..." the sentence should be "...the AICEB incorporates linear contrast..." lines 49-51: the sentence "Image enhancement algorithms can be classified into two main subsets: conventional methods that are manually constructed parameters, and deep learning-based methods for enhancing images." is not clear. I propose "Image enhancement algorithms can be classified into two main groups: conventional methods using manually constructed parameters and deep learning-based methods for improving images." line 57 : please change "...The gamma..." in "...the gamma..." line 142: please remove the first "image" word in the sentence line 143-144: what is the meaning of "the multi-scale image features U are first uniformly channelized", please clarify. line 147: there is a repetition of feature map symbol lines 169-170: I suggest to move the sentence "we segment the input feature channels and select only one-fourth of the total number of channels to extract and fuse features." in the section describing the global branch line 189: there is a repetition of "The global feature", please remove it. line 241-244: the sentence is a repetition of the previous sentence.
Response 2:
Thank you for your valuable comments. We have made changes in the corresponding positions you pointed out.
Reviewer 3 Report
Comments and Suggestions for Authors
(1) The writing format of the paper needs significant improvement, especially in the presentation of formulas and variables, which are highly non-standard.
(2) The paper should avoid using the first-person pronoun “we” in the description.
(3) The ablation study design is unreasonable. Although the authors compare the proposed method with current mainstream methods, they do not optimize the hyperparameters for each method. Therefore, the comparison of results is not valid.
(4) The selection of parameters for the three modules proposed in the paper appears to be arbitrary, lacking a proper optimization process.
Comments on the Quality of English LanguageThe English expression needs to be refined by a professional language editing service.
Author Response
Comments 1:The writing format of the paper needs significant improvement, especially in the presentation of formulas and variables, which are highly non-standard.
Response 1:
Thank you for your comments. We have standardized the presentation of formulas and variables in the manuscript.
Comments 2: The paper should avoid using the first-person pronoun “we” in the description.
Response 2:
Thank you for your comments. We have changed the way of expression in the article.
Comments 3:The ablation study design is unreasonable. Although the authors compare the proposed method with current mainstream methods, they do not optimize the hyperparameters for each method. Therefore, the comparison of results is not valid.
Response 3:
The ablation experiments in the manuscript are mainly used to verify the validity of the proposed module and the basis for the reasonable setting of the parameters in our method. As for the comparison with the current mainstream methods, we used an open implementation with default settings, which is a common practice in many previous works.
Comments 4:The selection of parameters for the three modules proposed in the paper appears to be arbitrary, lacking a proper optimization process.
Response 4:
This paper primarily proposes an adaptive image contrast enhancement module (AICEB). To this end, in-depth experimental analyses on the number of AICEBs in the model and the parameters governing adaptive termination within the module have been conducted through sensitivity tests. Regarding the optimization of the key parameters of our model, a cautious approach has been adopted.
Round 2
Reviewer 2 Report
Comments and Suggestions for Authors
No additional comments.
Reviewer 3 Report
Comments and Suggestions for Authors
The manuscript is acceptable for publication.